# Roles of Lytic Viral Replication and Co-Infections in the Oncogenesis and Immune Control of the Epstein–Barr Virus

**DOI:** 10.3390/cancers13092275

**Published:** 2021-05-10

**Authors:** Yun Deng, Christian Münz

**Affiliations:** Viral Immunobiology, Institute of Experimental Immunology, University of Zürich, 8057 Zürich, Switzerland; deng@immunology.uzh.ch

**Keywords:** CD8^+^ T cells, natural killer cells, CD27, humanized mice, malaria, human immunodeficiency virus (HIV), Kaposi sarcoma-associated herpesvirus (KSHV)

## Abstract

**Simple Summary:**

The Epstein–Barr virus (EBV) colonizes more than 95% of the adult human population. Its cancer-forming potential is usually contained by lifelong immune control. Genetic alterations and immune modulation by co-infection point towards cytotoxic lymphocytes, such as natural killer and CD8^+^ T cells, as the main pillars of this immune protection. In this review, we discuss how the EBV infection program that leads to infectious virion production and co-infections, such as with malaria parasites, the human immunodeficiency virus (HIV) and the Kaposi sarcoma-associated herpesvirus (KSHV), modulate this immune control.

**Abstract:**

Epstein–Barr virus (EBV) is the prototypic human tumor virus whose continuous lifelong immune control is required to prevent lymphomagenesis in the more than 90% of the human adult population that are healthy carriers of the virus. Here, we review recent evidence that this immune control has not only to target latent oncogenes, but also lytic replication of EBV. Furthermore, genetic variations identify the molecular machinery of cytotoxic lymphocytes as essential for this immune control and recent studies in mice with reconstituted human immune system components (humanized mice) have begun to provide insights into the mechanistic role of these molecules during EBV infection. Finally, EBV often does not act in isolation to cause disease. Some of EBV infection-modulating co-infections, including human immunodeficiency virus (HIV) and Kaposi sarcoma-associated herpesvirus (KSHV), have been modeled in humanized mice. These preclinical in vivo models for EBV infection, lymphomagenesis, and cell-mediated immune control do not only promise a better understanding of the biology of this human tumor virus, but also the possibility to explore vaccine candidates against it.

## 1. Introduction on EBV Infection and Oncogenesis

Epstein–Barr virus (EBV), also known as human herpesvirus 4 (HHV4), is one of the most successful human pathogens, with more than 95% of adults being persistently infected [1]. At the same time, it is the only human pathogen that can readily transform its primary host cell, human B cells, into immortalized lymphoblastoid cell lines (LCLs) in culture [2]. Indeed, EBV was also discovered in a human B cell malignancy, namely Burkitt’s lymphoma (BL), by Anthony Epstein and colleagues [3,4]. Therefore, the mystery that we are still trying to fully understand more than 50 years after the discovery of EBV is how the impressive oncogenic potential of EBV is controlled in most carriers despite persistent infection with this oncogenic γ-herpesvirus. 

Initially, it was assumed that the reduced viral oncogene expression that can be found in tumors of otherwise immunocompetent individuals is unique to these cancer settings. These infection programs are termed latency I, II, and III. Latency III shows the expression of all latent viral gene products—six nuclear antigens (EBNAs), two latent membrane proteins (LMPs), two non-translated small RNAs (EBERs), and more than 40 microRNAs (BHRF1 and BART miRNAs) [5,6]. In contrast, latency I, that can be found in BL, expresses only nuclear antigen 1 of EBV (EBNA1) at the protein level, and latency II, with the expression of EBNA1 and the two LMPs (LMP1 and 2), is present in Hodgkin’s lymphoma (HL) and nasopharyngeal carcinoma (NPC) [7]. EBERs and BART miRNAs are also expressed in these malignancies. While it is true that latency III lymphomas mainly occur during immune suppression, for example, after transplantation in post-transplant lymphoproliferative disorder (PTLD) or during HIV co-infection as immunoblastic lymphoma, rare cases of latency III lymphoma detected in diffuse large B cell lymphomas (DLBCLs) also occur in otherwise healthy older individuals [7]. Moreover, it was demonstrated by David Thorley-Lawson’s group that all tumor-associated latencies, including latencies I and II, are also present in healthy EBV carriers [8,9]. Moreover, these pre-malignant states might allow infected naïve B cells to transit into the memory B cell pool via providing signals for B cell survival in germinal centers and switching on EBNA1 expression for viral extrachromosomal DNA maintenance [10]. This allows EBV after transmission via saliva to differentiate B cells in submucosal secondary lymphoid tissues, like the tonsils, into the long-lived memory B cell pool and acquire in quiescent memory B cells true latency without protein expression (latency 0) in which only EBERs and BART miRNAs are present [11].

Under certain circumstances, primarily by recognizing the cognate antigen through the B cell receptor of infected cells, EBV can then reactivate into lytic replication from the latency 0 reservoir in memory B cells or from the BL-associated latency I infection program [12,13]. Accordingly, lytic EBV replication can be found in plasma cells of healthy EBV carriers that result from such B cell receptor-mediated activation and resulting differentiation [14]. Lytic replication, with its >80 proteins, has been divided into immediate early (IE), early (E), and late (L) gene expression based on the sequential order of detection. It is initiated by the expression of the plasma cell-associated transcription factors XBP1 and Blimp-1 that induce the expression of the IE proteins BZLF1 and BRLF1 [15,16]. BZLF1 and BRLF1 stimulate the expression of the E proteins, including BMRF1, BBLF2/3/4, BSLF1, BALF2, and BALF5 that form the viral DNA polymerase for EBV genome replication [17,18]. In addition to viral DNA replication [19,20], a viral pre-initiation complex (vPIC) including the E proteins BGLF4, BGLF3, BcRF1, BFRF2, and BVLF1 is required for L protein expression [20,21]. Only a few L genes do not require the vPIC and are under the direct control of BRLF1, such as the immune escape protein vIL-10 encoded by BCRF1 [22]. Most of the around 36 viral L gene products are structural proteins for the assembly of capsid, tegument, and envelope, such as BLLF1 (gp350/220) that attaches to CD21 on B cells [23]. They assemble virions for infectious EBV particle release from plasma cells. Basolateral infection of mucosal epithelia and lytic replication within them might then amplify EBV further for efficient shedding into the saliva for transmission [24]. 

Thus, all EBV infection programs, including those found in EBV-associated malignancies, are present in healthy carriers of the virus. Their transition into overt lymphomas needs to be immune controlled and we can learn how the human immune system can be taught to resist such a lifelong oncogenic threat by studying EBV-specific immune responses.

## 2. Role of Lytic Replication for EBV Infection, Immune Control, and Oncogenesis

### 2.1. Immune Control of EBV

Despite EBV’s contribution to 1.5% of all cancer cases worldwide, with an estimated 200,000 new cases reported every year [7,25], the majority of EBV infections are asymptomatic [1]. This delicate EBV–host balance is maintained by the anti-viral immune response within immunocompetent hosts. Although the host’s immune responses are insufficient to eliminate EBV, they provide immediate defense mechanisms during primary EBV infection, as well as subsequent immune control, that are important to maintain the infection in an asymptomatic state [26]. The main effector cell types involved in this immune control that will be discussed in this review include innate lymphocyte populations such as natural killer (NK) cells, γδ T cells, and NKT cells, as well as CD8^+^ and CD4^+^ T cells.

The function of NK cells in the EBV-specific immune control was described in patients with X-linked lymphoproliferative type 1 (XLP-1) syndrome caused by surface lymphocyte activation marker (SLAM)-associated protein (SAP) deficiency [27,28]. The regulation of SAP is key to the activation of NK cells through CD244 (2B4) and NTB-A co-stimulation [29,30]. NK cells that contain SAP mutations demonstrate a failure to kill EBV-infected LCLs [31]. This implies that the impaired cytotoxicity of NK cells may contribute to the pathogenesis of XLP-1 disease. Depletion of NK cells causes an increase in EBV viral loads and associated tumorigenesis in mice reconstituted with human immune system components (humanized mice) [32]. This underlines the direct contribution of NK cells to EBV-specific immune control. However, this was only seen in animals infected with wild-type EBV but not lytic replication-deficient BZLF1 knockout EBV, demonstrating a selective role of NK cells in restricting lytically EBV-infected cells in humanized mice [32]. Accordingly, B cells with lytic EBV infection are preferentially targeted by early differentiated NKG2A^+^KIR^−^ NK cells in vitro [33,34].

Besides NK cells, other innate cell populations, including Vγ9Vδ2 and NKT cells, can mount immune responses at different stages of EBV infection. Vγ9Vδ2 T cells induce cytotoxicity and cytokine responses by recognizing the NKG2D ligand ULBP4, which is upregulated upon EBV infection of B cells [35,36,37]. In addition, potent anti-tumor effects of Vγ9Vδ2 T cells against EBV-associated malignancies were observed in humanized mice reconstituted with EBV-infected human umbilical cord blood mononuclear cells (CBMCs) [38]. Interestingly, the expansion of the Vγ9Vδ2 T cells was seen in culture in response to only latent EBV^+^ Akata cells but not EBV^−^ Akata cells or lytic EBV^+^ Akata cells. Further experiments revealed that Vγ9Vδ2 T cells are preferentially stimulated by latency I-carrying BL cell lines that express EBNA1 as the sole viral protein, suggesting the importance of immune control of Vγ9Vδ2 T cells against the latency I stage of EBV infection [39]. Apart from Vγ9Vδ2 T cells, NKT cells expressing the semi-invariant Vα24-Jα18/Vβ11 T cell receptor recognize glycolipid antigens presented by the non-classical MHC class I-like molecule CD1d [40]. Since CD1d is downregulated on the fully transformed LCLs during latency III, NKT cells are suggested to be involved in regulating EBV latency II-infected cells [41]. In fact, only low NKT frequencies have been reported in EBV latency II HL and NPC [42]. Therefore, while NK cells primarily target lytically EBV-infected cells, Vγ9Vδ2 T cells and NKT cells seem to play a complementary role in regulating the EBV-infected cells at latency stages I and II, respectively.

In addition to innate cytotoxic lymphocytes, EBV infection elicits adaptive immune responses with a considerable expansion of virus-specific CD8^+^ T cells that can be detected in acute infectious mononucleosis (IM) blood samples using peptide-MHC class I tetramers [43,44]. In contrast, a small or no expansion in the CD4^+^ T cell compartment was seen in IM patient samples. Nevertheless, the detection of peptide-MHC class II tetramers for distinct EBV antigens indicates the presence of virus-specific CD4^+^ T cell populations [45,46]. Different responses to individual lytic and latent EBV epitopes have been described for CD8^+^ and CD4^+^ T cells. While early EBV antigen-specific CD8^+^ T cell responses showed immunodominance during acute infection, including epitopes of IE (BZLF1 and BRLF1) and E (e.g., BMLF1) proteins of lytic replication [35,39], CD8^+^ T cell responses to latent epitopes were found at a lower frequency with a focus on the family of nuclear antigens 3 of EBV (EBNA3A, 3B, and 3C) [47,48]. In contrast, CD4^+^ T cell responses are dominated by latent rather than lytic antigen recognition and display a broader immune response against different lytic proteins [49,50]. In the long-term memory response to lytic and latent antigens, CD8^+^ T cell responses present with a hierarchy of IE > E > L lytic antigen recognition and an increase in frequency for latent antigen detection [48,51]. A similar decrease in responses to lytic and latent epitopes from acute to persistent EBV infection was also seen for CD4^+^ T cells [26,52]. The most consistently recognized antigen for CD4^+^ T cells is EBNA1 [53,54]. Therefore, these findings demonstrate that both CD8^+^ and CD4^+^ T cells are activated during acute primary EBV infection, with CD8^+^ T cells preferentially targeting early lytic antigens, while CD4^+^ T cells show higher responses to latent antigens. The protective role of CD4^+^ and CD8^+^ T cells was demonstrated by antibody-mediated depletion or pharmacological T cell suppression in humanized mice which both increased EBV viral loads and lymphomagenesis [55,56,57,58,59]. Furthermore, adoptive transfer of EBV-specific T cells has been clinically used to successfully target some EBV-associated malignancies [60]. Thus, cytotoxic lymphocytes, including NK, NKT, Vγ9Vδ2 T, CD4^+^, and CD8^+^ T cells, contribute to lifelong immune control of persistent infection with human oncogenic EBV. 

### 2.2. Genetic Predisposition for Altered EBV Infection, Immune Control, and Oncogenesis

Primary immunodeficiency (PID) is a type of genetic predisposition which refers to genetic variations associated with disabled or absent functions of the immune system [61,62]. Patients with certain PIDs present with an increased susceptibility to EBV-associated diseases due to the lack of sufficient immune surveillance. Understanding the underlying disease mechanisms helps to identify molecules that are important for EBV-specific immune control.

PID studies reveal that genetic mutations in perforin, Munc 18-2 and 13-4 lead to chronic active EBV infection (CAEBV), as they compromise the cytotoxic effector machinery utilized mainly by T cells and NK cells [63,64]. Further studies showed that molecules that mediate the NK-B and T-B cell engagement, such as some (co-)stimulatory and (co-)inhibitory molecules, are highly relevant to the prevention of EBV-associated diseases. For instance, mutations in SH2D1/SAP [27,28], IL-2 inducible T cell kinase (ITK) [65], magnesium transporter 1 protein (MAGT1) [66], RASGRP1 [67,68], CD16 [69,70], CD27/CD70 [71,72,73,74], and the HLA-DRB1*15:01 haplotype [75] are associated with weakened EBV-specific immune control (Figure 1). Here, we will discuss some of these molecules and highlight recent research work on them. CD27 belongs to the tumor necrosis factor (TNF) receptor superfamily and is expressed on T cells [76], B cells [77], and NK cells [78]. CD70, the ligand for CD27, is expressed on subsets of dendritic cells (DCs) and activated lymphoid and myeloid cells [79,80]. Deficiencies in both CD27 and CD70 are associated with clinical symptoms of EBV^+^ lymphoproliferative diseases (LPDs), IM, and EBV viremia. The most prevalent EBV-associated lymphoma in those patients is HL, which occurs in 28% (9/33) of CD27-deficient patients and 44% (7/16) of CD70-deficient patients [81]. Mechanistically, it seems that the lytic antigen-specific CD8^+^ T cells (exemplified by early lytic antigen BMLF1) are primarily disrupted with impaired cell proliferation and cytotoxicity upon CD27 blocking in EBV-infected humanized mice [74]. In addition to PIDs, the genetic risk factor HLA-DRB1*15:01 (HLA-DR15) for multiple sclerosis (MS) conferred attenuated immune control of EBV infection which manifested with higher EBV viral loads in humanized mice [75]. Furthermore, HLA-DR15-restricted CD4^+^ T cell clones recognize LCLs less efficiently than their HLA-DRB1*04:01-restricted counterparts, suggesting an impaired killing capacity against EBV-infected B cells [75]. Therefore, these findings suggest that cytotoxic T cells are important for EBV-specific immune control. 

Although NK cell deficiencies are rare, genetic predispositions that affect the NK cell differentiation and stimulation can have profound effects on EBV-specific immune control. For instance, GATA-binding protein 2 (GATA2) has been described as regulating the development or maintenance of early-differentiated CD56^bright^ NK cells that are capable of cytotoxicity and cytokine production [82,83]. Patients diagnosed with GATA2 mutations showed higher susceptibility to EBV viremia, EBV-associated smooth muscle tumors (spindle cell tumors), and chronic active EBV (CAEBV) [84,85,86]. Similarly, defects in minichromosome maintenance complex component 4 (MCM4) also lead to disruption of NK cell development. Specifically, a decreased frequency in CD56^dim^ NK cells, as well as impaired proliferation and maturation in the CD56^bright^ population, has been seen in those patients [87,88]. This presumably increases the risk of developing EBV-associated lymphoproliferations [87]. Thus, mature cytotoxic NK cells (denoted by the CD56^bright^ subset) are essential to control EBV-associated lymphomas.

### 2.3. Influence of Lytic Replication on EBV Infection, Immune Control, and Oncogenesis

Although it has been commonly acknowledged that EBV^+^ tumors are largely composed of latently infected cells, accumulating evidence has shown a remarkable role of EBV lytic gene products in EBV-driven oncogenesis [1,89,90]. While a number of EBV lytic genes have been found to be expressed in a “leaky” fashion during the early stage of EBV infection, a very limited number of infectious virions were detected shortly after EBV entry into B cells, with reactivation usually not occurring for two weeks thereafter [91,92,93]. An abortive (incomplete) lytic replication cycle in the pre-latent stage is suggested as a first phase during the development of the EBV infection within the host [1,91,94,95]. This implies that the transient lytic gene expression may play a supportive role during latency development, tumor formation, and LCL generation. 

Even though the contribution of each individual EBV lytic gene to the tumorigenesis of EBV-associated malignancies remains unclear, lytic viral gene expression has been found to be associated with different types of EBV-positive tumors. BZLF1 is the principle transcriptional transactivator that switches EBV infection from latency to lytic replication. BZLF1 expression has been detected in various EBV-associated lymphomas, such as endemic BL [96,97], NPC [98,99], and gastric carcinoma [100]. Among patients with the same EBV-associated malignancy, however, different profiles of lytic gene expression were found. In one study of 12 endemic BL samples, lytic gene expression of BARF1, BHLF1, and BHRF1 was detected in diverse patterns among the eight tumor samples that were positive for BZLF1 expression [97]. Moreover, despite the detection of the lytic transactivator BZLF1, not all lytic genes were found in the respective tumor samples [100,101,102]. This might be due to certain types of tumors requiring lytic gene expression for survival or to change the tumor microenvironment through, e.g., immunomodulation. Nevertheless, these findings indicate that complete lytic replication is not required in EBV-driven lymphomas. 

The effect of lytic EBV replication for the oncogenesis of EBV has been extensively studied in humanized mice infected with either wild-type or lytic replication-deficient BZLF1 knockout EBV. In one of the studies, the NK population was depleted post EBV infection. Both the tumor incidence and EBV viral loads were elevated in the wild-type EBV-infected group but not in the BZLF1 knockout EBV-infected group [32]. This highlights the important role of NK cells in the regulation of EBV-associated malignancies due to their restriction of lytic viral replication. Another study investigated the role of CD27 in CD8^+^ T cells during EBV infection and unveiled different requirements for this co-stimulation in the recognition of lytic versus latent viral antigens. The blocking of CD27 had a more dramatic influence on the immune control of lytic EBV replication during wild-type EBV infection, as compared to BZLF1 knockout EBV infection that does not lead to lytic gene expression [74]. Even without compromised immune control, the direct comparison of wild-type and BZLF1-deficient EBV infection revealed slightly less lymphoma formation, especially in non-lymphoid tissues in the absence of lytic EBV reactivation [103]. Vice versa, enhanced BZLF1 expression or function leads to increased lymphoma formation in humanized mice [104,105]. Such increased but abortive early lytic EBV replication due to the loss of suppressive miRNAs or activating BZLF1 promotor polymorphisms is also observed in viral variants that are enriched in patients with EBV-associated malignancies [106,107]. Taken together, these findings reveal that lytic EBV reactivation is required for the pathogenesis of EBV-associated lymphomas.

## 3. Role of Co-Infections on EBV Infection, Immune Control, and Oncogenesis

Interestingly, such immune modulations that increase EBV-associated lymphomagenesis are often also associated with co-infections. The most famous example is probably the observation that endemic BL occurs at a high frequency in regions with holoendemic *Plasmodium falciparum* exposure, but not in regions where other malaria parasites are prevalent [108,109,110,111,112,113]. This association could result from compromised cell-mediated immunity against EBV and increased B cell stimulation by continuous malaria antigen exposure in the affected individuals [113] (Figure 2A). Along these lines, compromised EBV-specific T cell responses were observed in children of holoendemic malaria regions [114,115,116]. This might be a result of optimized parasite-controlling Th2 responses against the blood stage of *Plasmodium falciparum* [117], and, indeed, endemic BL development has been linked to well-controlled instead of pathogenic parasitemia [118]. Moreover, *Plasmodium falciparum* infection seems to drive NK cells to the dysfunctional CD56^−^CD16^+^ differentiation stage [119,120]. Therefore, the loss of immune control by both innate and adaptive cytotoxic lymphocytes might elevate the frequency of EBV-infected B cells from which BL develops, and indeed elevated EBV loads have been detected in continuously *Plasmodium falciparum*-exposed children [121,122,123]. Malaria antigen stimulation might then drive this increased pool of EBV-infected B cells into germinal center reactions from which this lymphoma is thought to emerge [111,113]. In germinal centers and by viral gene products, namely nuclear antigen 3C of EBV (EBNA3C), activation-induced cytidine deaminase (AID) is upregulated in B cells, facilitating the characteristic translocation of the oncogene c-myc into one of the immunoglobulin loci [124,125]. This translocation is characteristic for BL and allows the respective tumor cells to proliferate with the restricted EBV gene expression of the latency I program [111]. Thus, *Plasmodium falciparum* infection might support the development of BL by compromising EBV-specific immune control and driving infected B cells into a differentiation stage from which the respective tumor cells emerge.

An even more dramatic influence on EBV-specific immune control is shown by co-infection with human immunodeficiency virus (HIV) which elicits increased frequencies of EBV-associated lymphomas with mounting acquired immunodeficiency induced by this virus [126,127]. Less immunogenic EBV-associated lymphomas like Burkitt’s and HL are observed earlier during HIV infection, while highly immunogenic DLBCLs are found later in HIV infection. Accordingly, EBV with HIV co-infection in humanized mice compromises T cell-mediated immune control of EBV and results in elevated viral loads and lymphomagenesis [58]. Anti-retroviral therapy (ART) significantly decreased the incidence of highly immunogenic EBV-associated lymphomas in HIV-positive patients, but without dramatically influencing the occurrence of the less immunogenic tumors like BL and HL [126]. Therefore, HIV might play a role in EBV-associated lymphoma development beyond just suppressing T cell-mediated immune control. Along these lines, it was demonstrated that HIV can directly infect EBV-transformed B cells via CD4 that is up-regulated by EBV infection and CXCR4 that is maintained on the infected B cells [58] (Figure 2B). While these double-infected B cells were, however, efficiently immune controlled by CD8^+^ T cells due to upregulation of MHC class I-restricted antigen presentation [58], they might influence B cell physiology, EBV-specific T cell responses, or HIV latency reservoir composition and thereby facilitate the development of less immunogenic EBV-associated B cell lymphomas.

While EBV and HIV dual infection is less appreciated, another virus that co-exists with EBV in infected B cells is the Kaposi sarcoma-associated herpesvirus (KSHV) or human herpesvirus 8 (HHV8) [128,129,130,131]. This co-infection can be frequently found in primary effusion lymphoma (PEL) that is 100% associated with KSHV and, in the majority of cases (90%), also harbors EBV [127,132,133]. More recently, it was also found that EBV infection supports persistent KSHV infection both in humanized mice [131,134] and in vitro [130,135]. Consistent with these findings are the observations that KSHV is usually not found without EBV co-infection in African children [136,137], and that the monkey orthologs of these viruses can be co-transmitted in macaques [138]. Moreover, co-infection of EBV with KSHV in humanized mice causes increased lymphoma formation, and the emerging lymphomas display some features of plasma cell differentiation [131,134] which is characteristic for PELs [133]. These PEL-like tumors have an elevated reactivation of EBV to its lytic replication (Figure 2C). Co-infection with lytic replication-deficient BZLF1 knockout EBV and wild-type KSHV reduces lymphoma formation [131,134]. In addition, this co-infection drives NK cells into a dysfunctional CD56^−^CD16^+^CD38^+^CXCR6^+^ differentiation stage [134]. Thus, KSHV seems to enhance EBV-associated lymphomagenesis both by transactivating EBV lytic replication and possibly by immune modulation. 

Co-infection of EBV and *Helicobacter pylori* (*H. pylori*) has also been increasingly reported in gastric carcinoma (GC) patient samples with an increased severity of gastric inflammation, suggesting a crosstalk between the two types of infections [139,140]. Both EBV and *H. pylori* are well-known pathogens associated with various cancers and persist for life within the host. However, the underlying mechanism for the interaction between EBV and *H. pylori* is not fully understood. It was suggested that *H. pylori* induces the reactivation of the EBV lytic cycle in gastric epithelial cells due to the higher EBV viral loads found in *H. pylori*-infected samples compared to uninfected samples [141]. This is also supported by the detection of BZLF1 expression in gastric carcinoma [100]. Mechanistically, *H. pylori* enhances the EBV-induced proliferation by presumably altering the apoptotic pathway via p53 and Bax expression, and modulating these tumor suppressors with its cytotoxin-associated gene (CagA) protein [142,143,144]. However, due to the limited number of studies that have been published on EBV co-infection with *H. pylori*, further investigations into the regulation of EBV-associated pathology by *H. pylori* are needed.

## 4. Conclusions and Outlook

The discussed evidence suggests that EBV persists in healthy virus carriers with all of its infection programs that are also found in associated malignancies. These pre-malignant stages transit into lymphomas if EBV-specific immune control by primarily innate and adaptive cytotoxic lymphocytes is compromised. Recent evidence suggests that at least early abortive lytic EBV infection contributes to lymphomagenesis and that immune control of this lytic replication is necessary to prevent EBV-associated malignancies. Gene mutations or co-infections that compromise this immune control and HLA polymorphisms that render it less efficient are associated with the emergence of EBV-associated diseases. These can be modeled by blocking the molecules identified in PIDs with antibodies, choosing certain HLA haplotypes for reconstitution and HIV or KSHV co-infections in humanized mice. From these studies emerges a unique phenotype of cytotoxic lymphocyte differentiation that is essential for EBV-specific immune control and explains that mutations in essential molecules for this phenotype predispose in PIDs for EBV-associated pathologies and little else.

The challenge now becomes to stimulate as closely as possible this protective cytotoxic lymphocyte phenotype with vaccines to reinstate EBV-specific immune control in patients with virus-associated diseases and to establish these immune responses prior to delayed primary infection that often results in IM. Such vaccines could then be adapted to other tumor settings that require cytotoxic lymphocytes for remission.

Such vaccine candidates could then be tested in humanized mice which would be even more useful if we could model BL and HL in addition to DLBCL and PEL in this pre-clinical model. Along these lines, co-infection of EBV with *Plasmodium falciparum* or at least exposure to parasitic antigens and immune modulators might be explored. For the blood stage of malaria, however, the human erythrocyte reconstitution needs to be improved, while some features of the liver infection stage have been recently modeled in human hepatocyte-reconstituted mice [145,146,147]. Thus, due to the ubiquitous nature of persistent EBV infection, many molecular components have been identified that seem to be required for the immune control. Their corresponding mechanistic contribution to prevent EBV-associated malignancies can also be addressed more and more in humanized mice in loss-of-function experiments and these immunotypes can possibly be stimulated with vaccines in gain-of-function experiments. 

## Figures and Tables

**Figure 1 cancers-13-02275-f001:**
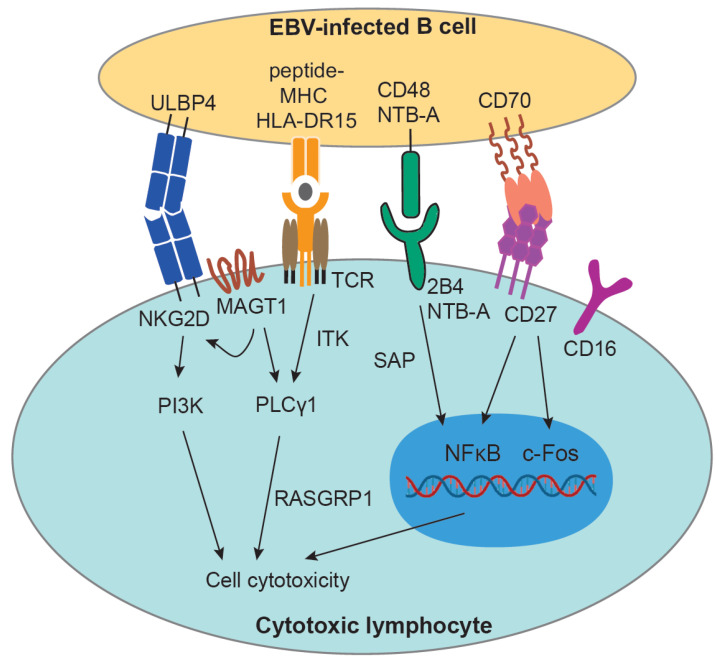
Signaling pathways involved in genetic predispositions that abolish the function of cytotoxic lymphocytes against EBV-infected cells. The graph summarizes genetic mutations that predispose for EBV-associated diseases: MAGT1 and ITK can both mediate the T cell activation by regulating PLCγ1. This pathway is further regulated by RASGRP1, which is the main activator for the Ras/MAPK signaling pathway for T and NK cell activation. In addition, MAGT1 also interacts with NKG2D to regulate the cytotoxic effector function of CD8^+^ T cells and NK cells. By engaging the receptors for 2B4 and NTB-A, they facilitate the NFκB signaling pathway through the interaction with the SAP adaptor protein. Similar for CD27, the engagement of CD70 leads to the activation of NF-κB and c-Fos/c-Jun pathways. Recent studies have also shown HLA-DR15 and CD16 as genetic risk factors for attenuated EBV immune control but the underlying mechanisms remain to be further investigated.

**Figure 2 cancers-13-02275-f002:**
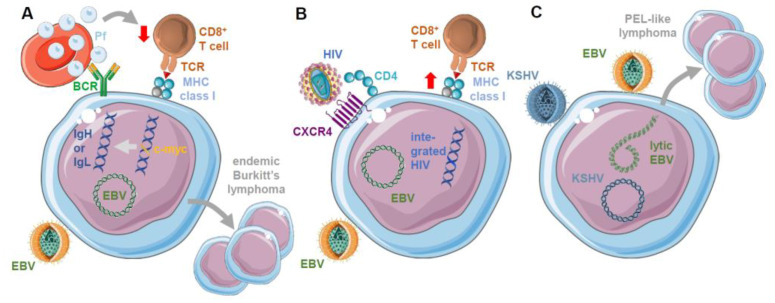
Co-infections by *Plasmodium falciparum* (Pf), human immunodeficiency virus (HIV), and Kaposi sarcoma-associated herpesvirus (KSHV) modulate Epstein–Barr virus (EBV) pathogenesis. (**A**) Holoendemic Pf infection is associated with the development of EBV-positive endemic BL. Pf is thought to drive EBV-infected B cells more frequently into the germinal center reaction where translocation of the oncogene c-myc into the heavy or light immunoglobulin chain (IgH or IgL) locus occurs that is characteristic for BL and responsible for its proliferation. In addition, Pf attenuates EBV-specific T cell-mediated immune control, thereby increasing the number of EBV-infected B cells. (**B**) HIV infection compromises EBV-specific immune control, but at the same time can also infect EBV-positive B cells that then are better controlled by CD8^+^ T cells due to an upregulation of MHC class I-restricted antigen presentation. (**C**) KSHV co-infection drives increased EBV-associated lymphomagenesis by transactivating lytic EBV replication in double infected B cells. This results in plasma cell differentiated lymphomas with similarities to primary effusion lymphoma (PEL). This figure was created in part with modified Servier Medical Art templates, which are licensed under a Creative Commons Attribution 3.0 unported license: https://smart.servier.com (accessed on 01 April 2021).

## Data Availability

Data sharing not applicable. No new data were created or analyzed in this study. Data sharing is not applicable to this article.

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
