# Peer review of "Roles of Lytic Viral Replication and Co-Infections in the Oncogenesis and Immune Control of the Epstein–Barr Virus"

_cancers, 2021, doi:10.3390/cancers13092275_

Round 1
Reviewer 1 Report
  This review describes the importance of EBV lytic infection and co-infection in tumorigenesis and immune response. Therefore, viral lytic and latent infections should be explained in the first half of the review, but there is very little explanation of lytic infections, which should be important for this paper. In addition, many EBV genomic defects have recently been discovered in EBV-infected lymphomas, and the link between abortive lytic replication and tumorigenesis has received a great deal of attention (Okuno Y et al, Nature Microbiol, 2019). However, abortive lytic replication is also poorly explained.
A similar pattern can be seen for co-infection. The author describes Malaria, HIV, and KSHV. However, other co-infections with various microorganisms such as H. pylori have also been reported, and these should be explained. In any case, the paper is inadequately organized according to the title, and several important papers are missing.
 The more fundamental problem is that there are a large number of grammatical errors, abbreviations, and misspellings in this paper. For example, since this virus was discovered by Dr. Epstein and Dr. Barr, most examples refer to it as Epstein-Barr virus. However, the title of this paper is Epstein Barr. In addition to these, there are a large number of other errors, and because of the number of errors, this paper does not reach the level of submission.
Author Response
We have now incorporated all of the reviewers’ suggestions and outline the respective changes in the point-by-point response below as well as by underlining in the revised manuscript version.
Reviewer #1
This review describes the importance of EBV lytic infection and co-infection in tumorigenesis and immune response. Therefore, viral lytic and latent infections should be explained in the first half of the review, but there is very little explanation of lytic infections, which should be important for this paper. In addition, many EBV genomic defects have recently been discovered in EBV-infected lymphomas, and the link between abortive lytic replication and tumorigenesis has received a great deal of attention (Okuno Y et al, Nature Microbiol, 2019). However, abortive lytic replication is also poorly explained.
We thank this reviewer for his/her suggestion. We have now described lytic EBV infection in more detail on page 2 and 3 of the revised manuscript version. Furthermore, we discuss now on page 7 of the revised manuscript that different profiles of lytic EBV gene expression can be found in tumors and argue for abortive lytic replication.
A similar pattern can be seen for co-infection. The author describes Malaria, HIV, and KSHV. However, other co-infections with various microorganisms such as H. pylori have also been reported, and these should be explained. In any case, the paper is inadequately organized according to the title, and several important papers are missing.
We have now discussed the evidence for an influence of Helicobacter pylori infection on EBV associated tumorigenesis on page 10 of the revised manuscript version.
The more fundamental problem is that there are a large number of grammatical errors, abbreviations, and misspellings in this paper. For example, since this virus was discovered by Dr. Epstein and Dr. Barr, most examples refer to it as Epstein-Barr virus. However, the title of this paper is Epstein Barr. In addition to these, there are a large number of other errors, and because of the number of errors, this paper does not reach the level of submission.
We apologize that some chapters of the manuscript contained grammatical errors. We have now significantly improved our text in too many locations to outline them individually.
Reviewer 2 Report
The manuscript by Y. Deng and C. Munz describe the relationship between immune control, EBV and oncogenesis. At the end also co-infections are taken into account. I believe that this review is well written, of interest and rather exhaustive. However I have few suggestions that in my opinion would help to improve it and to make more accessible to a broader audience.
- The first and introductory paragraph contains important and useful information. But I suggest reorganizing it to facilitate the reader. As it is now the life cycle of the virus is described through the first paragraph but it should be reconstructed by the reader from the different parts of the paragraph. I suggest re-organizing it as follows:
- Brief description of the viral cycle (first infection, abortive and productive cycle, latency and reactivation) and the genes involved and infection targets;
- Relationship between each latent phase with the different pathological conditions (tumor, PTLD).
In alternative it could be splitted in two, one paragraph for description of the EBV infection and one for situation where the different latent phases can be found.
- Even if it is not the topic of the review I believe it is worth mentioning more precisely the implication in oncogenesis of all the viral life cycle stages (first-infection, latency and reactivation). It would justify why the focus of the review is on the lytic cycle. It has been already extensively reviewed also by the authors (Münz, C. Nat. Rev. Microbiol.2019, Yin, H et al Med. Microbiol. Immunol. Berl. 2019; Raab-Traub Curr. Opin. Virol. 2012, Jha, H.C et al. Front. Microbiol. 2016, El-Sharkawy, Aet al. Front. Oncol. 2018, Germini et al 2020 Cancers) but I believe that it is worth mentioning as global overview that would help in understanding the topic of this review.
- In the paragraph “Role of lytic replication for EBV infection, immune control and oncogenesis” the acute infection and in some cases latency are mostly mentioned . There is no explicit or poor mention regarding the reactivation from latency which is an important aspect for PTLD. Moreover immunosuppression could be linked to EBV reactivation itself. I believe that it would be an important addition to the review that will cover all the aspect of the EBV infection, in particular of the lytic cycle (main topic of the review)
- As in each paragraph many information are provided I suggest to put a small phrase of conclusion, a “take home message” from each paragraph
Author Response
We have now incorporated all of the reviewers’ suggestions and outline the respective changes in the point-by-point response below as well as by underlining in the revised manuscript version.
Reviewer #2
The manuscript by Y. Deng and C. Munz describe the relationship between immune control, EBV and oncogenesis. At the end also co-infections are taken into account. I believe that this review is well written, of interest and rather exhaustive. However I have few suggestions that in my opinion would help to improve it and to make more accessible to a broader audience.
- The first and introductory paragraph contains important and useful information. But I suggest reorganizing it to facilitate the reader. As it is now the life cycle of the virus is described through the first paragraph but it should be reconstructed by the reader from the different parts of the paragraph. I suggest re-organizing it as follows:
Brief description of the viral cycle (first infection, abortive and productive cycle, latency and reactivation) and the genes involved and infection targets;
Relationship between each latent phase with the different pathological conditions (tumor, PTLD).
In alternative it could be splitted in two, one paragraph for description of the EBV infection and one for situation where the different latent phases can be found.
We agree with this reviewer and have now better structured the introduction into historical introduction, latency and tumors, lytic replication and presence of tumor associated EBV gene expression programs already in healthy EBV carriers.
- Even if it is not the topic of the review I believe it is worth mentioning more precisely the implication in oncogenesis of all the viral life cycle stages (first-infection, latency and reactivation). It would justify why the focus of the review is on the lytic cycle. It has been already extensively reviewed also by the authors (Münz, C. Nat. Rev. Microbiol.2019, Yin, H et al Med. Microbiol. Immunol. Berl. 2019; Raab-Traub Curr. Opin. Virol. 2012, Jha, H.C et al. Front. Microbiol. 2016, El-Sharkawy, Aet al. Front. Oncol. 2018, Germini et al 2020 Cancers) but I believe that it is worth mentioning as global overview that would help in understanding the topic of this review.
We have now modified the introduction to indicate that the oncogenic functions by EBV assist transition of infected cells into the memory B cell pool. This is now included on page 2 of the revised manuscript text.
- In the paragraph “Role of lytic replication for EBV infection, immune control and oncogenesis” the acute infection and in some cases latency are mostly mentioned . There is no explicit or poor mention regarding the reactivation from latency which is an important aspect for PTLD. Moreover immunosuppression could be linked to EBV reactivation itself. I believe that it would be an important addition to the review that will cover all the aspect of the EBV infection, in particular of the lytic cycle (main topic of the review)
Reactivation into lytic replication by plasma cell differentiation has now been incorporated into the more detailed description of lytic EBV replication on pages 2 and 3 of the revised manuscript text.
- As in each paragraph many information are provided I suggest to put a small phrase of conclusion, a “take home message” from each paragraph
Concluding sentences have now been added to the individual paragraphs.
Reviewer 3 Report
Roles of lytic replication and co-infections in the oncogenesis and immune control of the Epstein Barr virus by Deng and Münz is a comprehensive review of lytic replication factors associated with development of oncogenesis. It also examines the roles of co-infections (HIV, plasmodium falciparum, KSHV) as well. It is well organized, thorough and in ready to publish form. Figures are appropriate and explained well in the the legend. A discussion of the relationship of tegument proteins driving oncogenesis would have been an interesting addition to add, but is beyond the topic chosen.Author Response
We have now incorporated all of the reviewers’ suggestions and outline the respective changes in the point-by-point response below as well as by underlining in the revised manuscript version.
Reviewer #3
Roles of lytic replication and co-infections in the oncogenesis and immune control of the Epstein Barr virus by Deng and Münz is a comprehensive review of lytic replication factors associated with development of oncogenesis. It also examines the roles of co-infections (HIV, plasmodium falciparum, KSHV) as well. It is well organized, thorough and in ready to publish form. Figures are appropriate and explained well in the the legend. A discussion of the relationship of tegument proteins driving oncogenesis would have been an interesting addition to add, but is beyond the topic chosen.
We thank this author for his/her positive comments.
Round 2
Reviewer 1 Report
The resubmitted paper shows a lot of improvement over the previous version. However, there are still many problems with the English language and description. These problems can be solved by using the native checker or by reading the Author instruction carefully.
- The reference number and style used in this paper do not match the journal style. Please check all references and reference numbers using the examples.
Examples
N.K. Minkah, C. Schafer, and S.H.I. Kappe, Humanized Mouse Models for the Study of Human Malaria Parasite 815 Biology, Pathogenesis, and Immunity. Front Immunol 9 (2018) 807.
Minkah, N.K.; Schafer, C.; Kappe, S.H.I. Humanized Mouse Models for the Study of Human Malaria Parasite Biology, Pathogenesis, and Immunity. Front. Immunol. 2018, 9 , 807.
Despite EBV’s contribution to 1.5% of all cancer cases worldwide with an estimated 200,000 new cases reported every year [7; 25], the majority of EBV infections are asymptomatic [1].
Despite EBV’s contribution to 1.5% of all cancer cases worldwide with an estimated 200,000 new cases reported every year [7, 25], the majority of EBV infections are asymptomatic [1].
Vγ9Vδ2 T cells induce cytotoxicity and cytokine response by recognizing the NKG2D ligand ULBP4, which is upregulated upon EBV infection of B cells [35; 36; 37].
Vγ9Vδ2 T cells induce cytotoxicity and cytokine response by recognizing the NKG2D ligand ULBP4, which is upregulated upon EBV infection of B cells [35- 37].
- Unfortunately, there are still many grammatical errors. It is highly recommended to use high quality English checker before resubmitting.
2-1 How to use commas
 In general, when connecting three or more nouns with and, insert a comma before the and.
Example
These infection programs are termed latency 0, I, II and III. (Lane 42)
These infection programs are termed latency 0, I, II, and III. (Lane 42)
Many such errors are found in Lane 42, 69, 73, 75, 98, 145, 160, 176, 179, 181, and 239.
2-2  Mistakes in abbreviations, etc.
 Lane 149 changes Late to L. A similar mistake is found in Lane 213. The abbreviations BL, HL, and NPC are used for Burkitt's lymphoma, Hodgkin's lymphoma, and nasopharyngeal carcinoma, respectively, since they are listed repeatedly. In vitro, as described in Lane 1123, 337, should be written in italics.
Author Response
We have now incorporated all of the reviewers’ suggestions and outline the respective changes in the point-by-point response below.
Reviewer #1
The resubmitted paper shows a lot of improvement over the previous version. However, there are still many problems with the English language and description. These problems can be solved by using the native checker or by reading the Author instruction carefully.
We have now adjusted the referencing style and further improved the writing.
The reference number and style used in this paper do not match the journal style. Please check all references and reference numbers using the examples.
Examples
N.K. Minkah, C. Schafer, and S.H.I. Kappe, Humanized Mouse Models for the Study of Human Malaria Parasite 815 Biology, Pathogenesis, and Immunity. Front Immunol 9 (2018) 807.
Minkah, N.K.; Schafer, C.; Kappe, S.H.I. Humanized Mouse Models for the Study of Human Malaria Parasite Biology, Pathogenesis, and Immunity. Front. Immunol. 2018, 9 , 807.
Despite EBV’s contribution to 1.5% of all cancer cases worldwide with an estimated 200,000 new cases reported every year [7; 25], the majority of EBV infections are asymptomatic [1].
Despite EBV’s contribution to 1.5% of all cancer cases worldwide with an estimated 200,000 new cases reported every year [7, 25], the majority of EBV infections are asymptomatic [1].
Vγ9Vδ2 T cells induce cytotoxicity and cytokine response by recognizing the NKG2D ligand ULBP4, which is upregulated upon EBV infection of B cells [35; 36; 37].
Vγ9Vδ2 T cells induce cytotoxicity and cytokine response by recognizing the NKG2D ligand ULBP4, which is upregulated upon EBV infection of B cells [35- 37].
References and call-outs in the text have been reformatted.
Unfortunately, there are still many grammatical errors. It is highly recommended to use high quality English checker before resubmitting.
2-1 How to use commas
In general, when connecting three or more nouns with and, insert a comma before the and.
Example
These infection programs are termed latency 0, I, II and III. (Lane 42)
These infection programs are termed latency 0, I, II, and III. (Lane 42)
Many such errors are found in Lane 42, 69, 73, 75, 98, 145, 160, 176, 179, 181, and 239.
The grammatical errors have been corrected.
2-2  Mistakes in abbreviations, etc.
Lane 149 changes Late to L. A similar mistake is found in Lane 213. The abbreviations BL, HL, and NPC are used for Burkitt's lymphoma, Hodgkin's lymphoma, and nasopharyngeal carcinoma, respectively, since they are listed repeatedly. In vitro, as described in Lane 1123, 337, should be written in italics.
The respective abbreviations have been introduced.